# Erosion–Corrosion Behavior of Friction Stud Welded Joints of X65 Pipelines in Simulated Seawater under Different Flow Rates

**DOI:** 10.3390/ma16124326

**Published:** 2023-06-12

**Authors:** Jie Zhao, Yuqi Feng, Hui Gao, Lei Wang, Xiaoyu Yang, Yanhong Gu

**Affiliations:** 1School of Safety Engineering, Beijing Institute of Petrochemical Technology, Beijing 102617, China; 2School of Mechanical Engineering, Beijing Institute of Petrochemical Technology, Beijing 102617, China; fengyuqi@bipt.edu.cn (Y.F.); guyanhong@bipt.edu.cn (Y.G.)

**Keywords:** erosion–corrosion, flow rate, X65 pipeline steel, friction stud welded joints

## Abstract

In order to study the complex erosion–corrosion mechanism of friction stud welded joints in seawater, experiments were carried out in the mixed solution of 3 wt% sea sand and 3.5% NaCl at flow rates of 0 m/s, 0.2 m/s, 0.4 m/s, and 0.6 m/s. The effects of corrosion and erosion–corrosion at different flow rates on materials were compared. The corrosion resistance of X65 friction stud welded joint was studied by electrochemical impedance spectroscopy (EIS) and potentiodynamic polarization (PDP) curves. The corrosion morphology was observed by a scanning electron microscope (SEM), and the corrosion products were analyzed by energy dispersive spectroscopy (EDS) and X-ray diffraction (XRD). The results showed that the corrosion current density decreased first and then increased with the increase in the simulated seawater flow rate, which indicated that the corrosion resistance of the friction stud welded joint increased first and then decreased. The corrosion products are FeOOH (α-FeOOH and γ-FeOOH), and Fe_3_O_4_. According to the experimental results, the erosion–corrosion mechanism of friction stud welded joints in seawater environment was predicted.

## 1. Introduction

Friction stud welding was invented by the British Welding Institute in the 1980s. Due to the growth of the offshore oil industry, friction stud welding has become a popular method for welding underwater pipelines. This technique is not affected by high-pressure environments and produces high-quality weld joints [1,2]. Considerable research has been conducted on the mechanical properties, technological processes, and metallographic changes of friction stud welded joints. However, due to their relatively weak corrosion resistance, there is a growing need to investigate the corrosion behavior of these welded parts [2,3,4].

Ma [4] employed friction stud welding to join the stud with X65 steel, and investigated the interrelation among microstructure, microhardness, chemical composition, and local electrochemical behavior in a 3.5% NaCl simulated seawater solution. Ma [2] innovatively studied the localized corrosion behavior of the joint surface and coating surface defects using a microarea electrochemical scanning system in a 3.5% NaCl simulated seawater solution. The findings indicate that the corrosion resistance at the weld seam is superior to other areas. The prior research has primarily investigated the static corrosion behavior of friction stud welded joints. However, the seawater environment is complex and variable, as demonstrated by Wang’s [5] study on circulation in the middle and deep layers of the South China Sea, where the current velocity ranges from 0 m/s to 0.6 m/s in some areas. The erosion–corrosion of welded joints in seawater is a synergistic effect of the electrochemical mechanism and mechanical mechanism [6]. Hence, studying the erosion–corrosion of friction stud welded joints is more representative of their real-world application in oceanic environments compared to static corrosion studies [7]. However, at present, there are few studies on the mechanism of erosion–corrosion about the friction stud welding joint [8].

The flow velocity, concentration of sea sand, and impact energy are three primary factors influencing erosion, with the latter two being governed by the flow velocity [6,9,10,11,12]. Thus, investigating the effect of flow velocity on erosion–corrosion is crucial for understanding the mechanism of such degradation [13]. Luo [14] utilized an electrochemical method to investigate the correlation between the flow rate and erosion, concluding that the flow rate influenced both the cathodic and anodic processes of corrosion. Zheng [6] examined the effects of the flow rate and water pressure on the erosion of X65 steel, and found that the erosion mechanism differed from that of static corrosion due to the flow rate. Erosion–corrosion is a dynamic process influenced by frequent fluid impacts, emphasizing the need for dynamic electrochemical measurements [15]. In a study by Xu [16], a flowing hydrodynamic environment was simulated to investigate the initiation and dynamic progression of erosion–corrosion on a pre-corroded X65 pipeline steel surface, utilizing coupon electrodes and wire beam electrodes. The results indicated that the X65 pipeline steel suffered more severe erosion damage when already corroded on its surface. Parancheerivilakkathil [17] explored the factors influencing jet erosion and concluded that the mass loss increases with flow velocity [18]. Elemuren [19] suggested that synergistic effects are dominant at low flow velocities, whereas erosion becomes the primary factor at high flow velocities. The research found that erosion–corrosion of pipeline steel in seawater undergoes two stages with changes in flow velocity: a synergistic effect of erosion and corrosion, and a dominant effect of erosion. Currently, there is limited research on the direct measurement of the dynamic process of erosion–corrosion under different flow rates using electrochemical methods, as well as the comparison between dynamic and static erosion–corrosion.

To investigate the corrosion mechanism of X65 friction stud welded joints in seawater, the study examined the corrosion behavior of the joint under different conditions: static corrosion and flow rates of 0.2 m/s, 0.4 m/s, and 0.6 m/s. The experiment used a simulated seawater environment test method and a dynamic simulation electrochemical reactor test platform. The corrosion current and impedance were measured by dynamic electrochemical method. The corrosion morphology was observed by a scanning electron microscope (SEM), and the three-dimensional corrosion morphology was measured by the depth of field (DOF). The corrosion products were analyzed by energy dispersive spectroscopy (EDS) and X-ray diffraction (XRD). The erosion–corrosion mechanism of an X65 friction stud welded joint in a complex seawater environment is proposed.

## 2. Experimental

### 2.1. Material and Sample Preparation

The experimental welding joints are made of 16Mn steel and X65 steel base material. X65 pipeline steel is used as the welding base material in the welding process. The chemical composition of X65 (calculated by mass fraction) is as follows: C 0.09%, Si 0.26%, Mn 1.29%, P 0.012%, S 0.004%, Cr 0.07%, Ni 0.15%, Mo 0.09%, and Fe allowance. The stud material is 16Mn steel, the chemical composition of 16Mn (calculated by mass fraction) is: C 0.13–0.19%, Mn 1.2–1.6%, S less than 0.03%, Cu less than 0.25%, Cr less than 0.3%, and Ni less than 0.3%. The underwater friction stud welding (U-FSW) welding process was used to carry out welding tests in simulated seawater to obtain the welded sample, and then the wire cutting was carried out from the middle part of the weldment.

The specimen was processed into a disc with a diameter of 10 mm and a thickness of 2 mm. The final joint sample retained four parts: the stud zone (SZ, stud zone), heat affected zone (HAZ, heat affected zone), weld zone (WZ, welded zone), and base metal zone (BMZ, base metal zone). The specific location distribution is shown in Figure 1; it can be seen that the FSW welded joint receives a concentrated high heat, which makes the weld zone more dense. After grinding and polishing the working face, it was cleaned with deionized water, dehydrated with ethanol or acetone, dried with cold air, and dried in reserve. The non-working face is connected with the working electrode clamp. In Figure 1, 1, 2, and 3 are the three observation points of SEM, and a, b, c, d represent specimens with a flow velocity of 0 m/s, 0.2 m/s, 0.4 m/s, and 0.6 m/s, respectively.

To simulate the conditions of the seabed, NaCl was completely dissolved by stirring. As for the mixing of sea sand, at low flow rates, most of the sea sand settles at the bottom and only a small amount of fine particles are carried along with the flow. Since sea sand on the seabed is not uniformly distributed, it was mixed and stirred in this experiment to deposit the sand at the bottom, preparing for experiments at different flow rates. A 3.5% NaCl solution was selected as the experimental solution, with the addition of sea sand particles with sizes of 0.25 mm, 0.5 mm, and 1.0 mm to simulate their presence in seawater [2]. The mass fraction of sea sand used in the experiment was 3.5 wt%, and the experimental pressure was 1 MPa at room temperature. The oxygen content at the seabed is typically lower than that of the surface of seawater, and decreases with increasing depth. The exact amount of oxygen can vary depending on factors such as location, season, and depth. In general, the oxygen content in seabed water is relatively low, usually around 5–7 mg/L of dissolved oxygen. This study ensured the presence of oxygen in simulated seawater under a certain pressure, which is closer to the service environment of pipeline steel.

### 2.2. Electrochemical Measurements

Figure 2 depicts the electrochemical measurement system used in this study. The system includes a high-power electrochemical workstation from Princeton (Versastat 3F, AMETEK, Berwyn, PA, USA) and a self-developed high-temperature and high-pressure mechanical stirring electrochemical reactor used for corrosion testing. The classical three electrode system is used for electrochemical test, and the auxiliary electrode is a 10 mm × 10 mm platinum sheet, an Ag/AgCl electrode is used as the reference electrode, and the working electrode is a disc with a diameter of 10 mm and a thickness of 2 mm. The open circuit potential (OCP) was stable in about 30 min; the scanning range of electrochemical impedance spectroscopy (EIS) is 100 kHz to 10 mHz, plus there is a sine wave amplitude of 10 mV; the test range of potentiodynamic polarization curve (Tafel) is from −250 mV to 500 mV relative to the OCP, and the scanning rate is 0.5 mV/s. The polarization curves are fitted with the Tafel method, and the EIS data are fitted with ZsimpWin 3.50 software. EIS and Tafel experimental data were collected for each sample after 1 h and 168 h of immersion.

### 2.3. Surface Characteristics

The eclipse MA200 stereomicroscope from Nikon company in Tokyo, Japan was used to photograph the macro morphology after corrosion. The surface structure (SEM) of the coating was analyzed by an ssx-550 scanning electron microscope from the Shimadzu company, Tokyo, Japan. The measuring point position of SEM on the test piece is shown in Figure 1. The three measuring points 1, 2, and 3 cover the welded joint, weld, and heat affected zone. Symbols (a, b, c, d) are the micro morphology of welded joints immersed in 3.5% NaCl solution for 168 h at flow rates of 0 m/s (a), 0.2 m/s (b), 0.4 m/s (c), and 0.6 m/s (d). The phase composition and corrosion products of the coating were analyzed by a D8 advance X-ray diffractometer (XRD) from the Brooke AXS company (Karlsruhe, Germany). The scanning angle range was 10°~80°, the scanning step angle was 0.05°, and the scanning rate was 5°/min.

## 3. Result and Discussion

### 3.1. Electrochemical Characterization

#### 3.1.1. Analysis of EIS

The Bode diagram of the X65 friction stud welded joint immersed in 3.5% NaCl solution for 1 h and 168 h under simulated real working conditions is shown in Figure 3. By comparing the impedance values at 0.02 Hz at various flow rates, it can be seen that under static corrosion conditions (0 m/s), with the increase in immersion time, the impedance value increases, and the corrosion products have a certain degree of protection. When the flow rate is 0.2 m/s, the impedance value of the specimen increased significantly with the increase in immersion time, but decreased when the flow rate was 0.4 m/s and 0.6 m/s. It can be seen from Figure 3a that the impedance value of the specimen increases with the increase in flow rate after 1 h under the coupling effect of the corrosion and erosion. As can be seen from Figure 3b, the impedance value of the test piece decreases with the increase in the flow rate after 168 h of dynamic experiment. It can be considered that when corrosion begins, the increase in flow rate will reduce the corrosion rate. After a period of corrosion, the increase in flow rate will accelerate the corrosion rate. Comparing the impedance values of the test piece for 1 h and 168 h at the same flow rate, it can be seen that the impedance of the test piece for 168 h is greater than that after 1 h at the flow rate of 0.2 m/s and static corrosion conditions. On the contrary, when the flow velocity is 0.4 m/s and 0.6 m/s, the impedance value of the specimen at 168 h is less than that at 1 h. When the flow rate is less than 0.2 m/s, the corrosion product film is in the state of accumulation, and the compactness and protection of the corrosion products are improved. When the flow rate is greater than 0.4 m/s, the corrosion products are stripped after impact, and the compactness and protection of the corrosion product film are reduced.

The Nyquist diagram and local enlarged diagram of the high-frequency region obtained from the experiment are shown in Figure 4. After soaking the specimen for 1 h, the radius of the capacitive reactance arc increases with the increase in flow velocity, and the impedance of the deposition and diffusion in the working electrode increases, and the high-frequency region is controlled by dynamics. On the working electrode, the low-frequency impedance is mainly controlled by the mass transfer, and the high-frequency impedance is mainly controlled by the charge transfer [20]. In the low-frequency region, due to the mass transfer control, the electrode impedance spectrum is a single capacitive reactance arc with a large impedance [15,21,22,23], and there is a straight-line segment in the low-frequency region, as shown in Figure 4(a1).

As shown in Figure 4(b1), after the X65 friction stud welded joint is immersed in 3.5% NaCl solution for 168 h under simulated real working conditions, the capacitive arc radius first increases and then decreases with the increase in flow rate. This change law is the same as the experimental law of Zheng T G [24] and Zheng Q [6]. Under the condition of the low flow rate, the reaction film produced by corrosion is not easy to fall off, and the corrosion products have a protective effect on the substrate [14]. The flow rate of 0.2 m/s is more suitable for the accumulation process of the corrosion product film, so that the corrosion product film produced at the flow rate of 0.2 m/s is more stable than other groups of flow rates, the radius of the capacitive reactance arc is greater than the other flow velocities. When the flow rate is higher than 0.2 m/s, the increase in liquid flow rate leads to the increase in the joint surface shear stress and the reaction mass transfer rate, which is the main reason for the increase in surface electrochemical reaction rate. With the continuous increase in the flow rate, the electrode surface of X65 friction stud welded joint subjected to strong hydrodynamics will change its conventional dissolution rate and dissolution mechanism, and the surface is accompanied by pit and point local corrosion. With the increase in hydrodynamics, the impedance decreases significantly. The erosion is mainly affected by the mass transfer rate and wall reaction process.

As shown in Figure 4(a2,b2), the X65 friction stud weld joint produced a layer of high-frequency capacitive arcs in the high-frequency region after 168 h of immersion in 3.5% NaCl solution compared with the local magnification of the Nyquist high-frequency region after 1 h of immersion. In the electrochemical impedance spectrum, there are two capacitive reactance arcs, where the high-frequency capacitive reactance arc originates from the surface of the X65 friction stud welded joint, representing the film information, and the low-frequency capacitive reactance arc originates from the new interface exposed after denudation and in contact with the corrosive medium. Due to the increase in immersion time and the long-time contact between time and solution, the ion migration in the solution is no longer controlled by the external rate of motion. The fitting circuit soaked for 168 h has no Warburg impedance, and the fitting circuit soaked for 1 h and 168 h is only the difference between the Warburg impedance, indicating that soaking for 1 h and 168 h will not change the corrosion mechanism of the joint [17].

ZsimpWin software was used for the EIS parameter fitting, and the equivalent circuit diagram is shown in Figure 5. As shown in Figure 5a, the EIS results of friction stud welded joints soaked for 1 h at dynamic flow rate of 0.2 m/s are fitted with an {R (C (R (C (RW))))} equivalent circuit. As shown in Figure 5b, the EIS results soaked for 168 h are fitted with an {R (C (R (Q (R (QR))))} equivalent circuit. The EIS results of soaking for 1 h are also fitted with an {R (C (R (C (RW))))} equivalent circuit. As shown in Figure 5c, the EIS results of welding joints soaked for 168 h under dynamic flow rate (except 0.2 m/s) are fitted with an {R (C (R (QR)))} equivalent circuit.

C_dl_ and Q_dl_ are the capacitances of the double layer capacitance; C_po_ is the capacitance of the corrosion product film (Fe_2_O_3_ and γ-FeOOH); R_ct_ is the charge transfer resistance; R_po_ is the resistance of the corrosion product film and R_s_ is the solution resistance; Q_po_ is the capacitance of the double charge layer of the second corrosion product film (Fe_3_O_4_ and α-FeOOH) at the flow rate of 0.2 m/s; R_po_′ and R_po_″ are the resistance of the corrosion product film at the flow rate of 0.2 m/s; and Z_W_ is the Weber impedance [25].

Table 1 lists the fitted parameters of the EIS results measured after the immersion of the X65 friction stud welded joint in a 3.5% NaCl sand solution for 1 h and 168 h under simulated actual working conditions. At different flow rates, the radius of Nyquist capacitive reactance arc shows the same law as the sum of R_s_ and the polarization resistance (R_ct_ + R_po_). Under the simulated actual working conditions, dynamic erosion–corrosion coupling, and static conditions, the conductivity of the solution is good, so the value of solution resistance R_s_ is low. After soaking for 168 h, the solution resistance Rs showed a trend of first increasing and then decreasing with the increase in flow rate. It reached its peak at a seawater velocity of 0.4 m/s and reached its minimum at 0.6 m/s. It indicates that the corrosion products have a certain degree of protection, and at a flow rate of 0.6 m/s, the corrosion products are peeled off.

#### 3.1.2. Analysis of PDP

The corrosion current density I_corr_ is fitted by the Tafel fitting method, and the fitting results are shown in Table 2. Figure 6a,b show the polarization curves of the test piece after 1 h and 168 h under the coupling effect of the static corrosion and erosion–corrosion, respectively. The corrosion current density of the specimen decreases with the time at the static state (0 m/s) and at a flow rate of 0.2 m/s, while it increases with the time at 0.4 m/s and 0.6 m/s. Under the condition of erosion–corrosion, the self-corrosion current density of the specimen for 1 h decreases from 120.31 μA/cm^2^ to 35.42 μA/cm^2^ with the increase in the flow rate, indicating that the increase in the flow rate will reduce the corrosion rate when the corrosion product film has not been fully formed. After 168 h under corrosion erosion condition, the self-corrosion current density of the specimen increased from 20.39 μA/cm^2^ to 84.22 μA/cm^2^ with the increase in the flow rate. This phenomenon shows that after the corrosion product film is completely formed, the increase in the flow rate will lead to the increase in the corrosion rate [10,21,26]. After 168 h, the self-corrosion current density of the specimen under static corrosion is greater than that when the flow rate is 0.2 m/s and less than that when the flow rate is 0.4 m/s. This phenomenon shows that the corrosion rate of the specimen is the smallest when the flow rate is 0.2 m/s. The rule is consistent with the results of the impedance study.

### 3.2. Corrosion Morphology Analysis

Figure 7 shows the macro morphology of the specimen after 168 h of experiment. In Figure 7a, the surface of the test piece is covered by black corrosion products under static corrosion. When the flow rate of the specimen in Figure 7b is 0.2 m/s, the surface of the corrosion products is yellow and reddish brown. When the flow rate of the specimen in Figure 7c is 0.4 m/s, some corrosion products fall off, and some reddish brown and black corrosion products can be seen on the surface of the specimen. When the flow rate of the specimen in Figure 7d is 0.6 m/s, only a small part of reddish-brown corrosion products are left on the surface, which can clearly distinguish the four areas of the specimen, with metallic luster and erosion gullies.

Comparing Figure 7a–d, it can be seen that the composition of the corrosion products on the static corrosion surface is different from that on the surface under the coupling action of corrosion and erosion. The corrosion products on the surface of the specimen under a static state are mainly black substances, while the surface of the specimen under corrosion erosion is mainly yellow and reddish-brown substances. By comparing the surface macro morphology of the specimens at different flow rates under corrosion erosion, it can be seen that the severity of corrosion product film flaking increases gradually with the increase in flow rate [18]. The electrochemical experiment results show that the corrosion product film of the test piece can play a certain protective role, but the increase in the flow rate will cause the corrosion product film to fall off and the corrosion rate will be improved [22]. This macroscopic morphology can explain the conclusion of electrochemical experiment on the relationship between the flow rate and corrosion rate to a certain extent.

As shown in Figure 8, the welded joints of the X65 friction stud under simulated real working conditions are the micro morphology of 0 m/s (a1~a3), 0.2 m/s (b1~b3), 0.4 m/s (c1~c3), and 0.6 m/s (d1~d3) specimens after 168 h of experiments (the detection position show as in Figure 1). It can be seen from Figure 8(a1–a3) that the test piece is in a state of uniform corrosion. Static corrosion (0 m/s) products are mainly composed of a needle and flake structure, and the corrosion product film is relatively loose and accompanied by cracks [27].

It can be seen from Figure 8(b1–b3) that when the flow rate is 0.2 m/s, the corrosion product film surface of the specimen under the combined action of corrosion and erosion is mostly spherical and a cotton floc structure, and the surface is relatively sparse [26]. It can be seen that, different from the static corrosion test piece, the corrosion product film of the test piece is more uneven. It can be clearly seen that some surface corrosion products fall off and the remaining bulges and depressions are caused by impact. It can be seen from Figure 8(c1–c3) that when the flow rate is 0.4 m/s, the surface of corrosion product film of the test piece is dense under the joint action of corrosion and erosion. There is a phenomenon of accumulation on the surface of the corrosion products in the weld area in Figure 8(c2) of the test piece. It can be seen from Figure 8(d1–d3) that when the flow rate is 0.6 m/s, the corrosion product film of the specimen is almost completely stripped under the combined action of corrosion and erosion. The erosion grooves consisting of pitting pits can be clearly seen in the figure, and the surface of the other parts is relatively flat. In contrast, when the flow rate is 0.2 m/s, there is a small part of stripping and more accumulation of the corrosion products on the basis of static corrosion, resulting in the thickening of corrosion product film. When the flow rate continues to increase, the corrosion products peel off more until the material matrix and erosion ditch are exposed. This phenomenon is consistent with the conclusion of macroscopic morphology and the result of the relationship between the corrosion rate and flow rate in electrochemical experiments.

Figure 9 shows the depth of field morphology of the X65 friction stud welded joint immersed in a 3.5% NaCl solution for 168 h at different positions of 0 m/s (a), 0.2 m/s (b), 0.4 m/s (c), and 0.6 m/s (d) under simulated real working conditions. It can be seen from Figure 9a that the corrosion product film is uneven and has obvious pits under the action of static corrosion (0 m/s), and the maximum longitudinal height difference is 112.5 μm. The appearance of the corrosion product film is black, and the pit position is brownish yellow. The corrosion product film in Figure 9b still has some pits, but the maximum longitudinal height difference is reduced to 68.3 μm. The brownish-yellow corrosion products in the corrosion products increase, and only a small amount of black corrosion products can be seen. This change shows that the erosion of seawater will lead to the falling off of the corrosion product film and the flattening of the corrosion product film. It can be seen from Figure 9c that the corrosion product film is flat. There is no obvious corrosion pit that can be seen, and the maximum reduction in longitudinal height difference is 44.8 μm. It can be seen that the bottom layer of the corrosion product film is black and the surface layer is brownish yellow. As Figure 9d shows, there is an obvious erosion ditch, and the corrosion product film is almost completely stripped, exposing a bright white matrix. As the flow rate increases, the height of corrosion products shows a trend of first decreasing and then increasing. When the flow rate is at 0 m/s, the accumulation of corrosion products gradually increases the height, until the flow rate gradually increases to 0.4 m/s. After the corrosion products are eroded and peeled off, the height gradually decreases. When the flow velocity reaches 0.6 m/s, the erosion ditch is exposed, resulting in an increase in the overall height difference. The above phenomena show that the impact of seawater makes the corrosion products fall off, and the greater the flow rate, the more the corrosion products fall off. Moreover, the increase in the flow rate can accelerate the material transfer, the protective effect of corrosion product film is weakened, and the corrosion rate will increase with the increase in flow rate.

### 3.3. Corrosion Product Analysis

Figure 10 shows the surface mass percentage of the test piece tested with EDS after 168 h of immersion under various conditions. According to the EDS results, the corrosion products are mainly composed of Fe and O, and some salts precipitated from the solution containing 3.5% NaCl, such as Na^+^ and Cl^−^, are adsorbed. The test results show that trace elements such as Mn and Cr mainly come from the X65 friction stud welded joint. The contents of the elements are different under various conditions, but the types are the same [18,25,28,29].

The EDS elemental composition is mainly composed of Fe and O, and the corresponding corrosion products are Fe_2_O_3_, FeOOH, and Fe_3_O_4_ [30,31]. Due to the existence of chloridion in the solution, it is easy to destroy the corrosion product film in the corrosion system, which intensifies the risk of corrosion. After the corrosion products are spalled off, the matrix is exposed to the corrosion environment and continues to aggravate the corrosion. In addition, the effect of erosion can make Fe_2_O_3_, FeOOH, and Fe_3_O_4_ on the surface of welded joints susceptible to flaking.

Figure 11 shows the XRD spectrum of the X65 friction stud welded joint after immersion in simulated seawater at different flow rates for 168 h under simulated real working conditions. According to the results of elements contained in the corrosion products detected by EDS, the classification of the corrosion products under various conditions is analyzed in combination with jade peak seeking software (Jade 5), as shown in Figure 11. Under static corrosion (0 m/s), the XRD results show that the corrosion products on the surface of the specimen after immersion for 168 h are mainly α-FeOOH, γ-FeOOH, Fe_3_O_4_, and Fe_2_O_3_ peaks. When the flow rate is 0.2 m/s, the composition of the corrosion products is similar to that of static corrosion. However, when the flow rate rises to 0.4 m/s and 0.6 m/s, there is only γ-FeOOH and a small amount of Fe_2_O_3_. It can be seen that when the flow rate is 0.4 m/s and 0.6 m/s, the original corrosion product film is stripped. At this point, the metal matrix continues to corrode as an anode forming two initial corrosion products, γ-FeOOH and Fe_2_O_3_. The formation rate of these two corrosion products is slightly higher than the stripping rate, so only the peaks of these two corrosion products can be seen in the XRD results [25].

### 3.4. Erosion–Corrosion Mechanism

By summarizing the analysis results of the electrochemical experiment, SEM, EDS, and XRD, the erosion–corrosion mechanism of the X65 friction stud welded joint in simulated seawater is shown in Figure 12. As shown in Figure 12a, under static corrosion (0 m/s), the samples were electrochemically reacted to form Fe(OH)_2_.

The reaction is as follows:Anodic reaction: Fe − 2e^−^→Fe^2+^(1)

The cathode’s response process is as follows: O_2_ + H_2_O + 4e^−^→4(OH)^−^(2)
Fe^2+^ + 2OH^−^→Fe(OH)_2_(3)

They produce γ-FeOOH and Fe_2_O_3_ when the dissolved oxygen is sufficient. Fe_2_O_3_ is formed by the dehydration of Fe(OH)_3_ (refer to Figure 11).
4Fe(OH)_2_ + O_2_ + 2H_2_O→4Fe(OH)_3_(4)
Fe(OH)_3_→FeOOH + H_2_O(5)
2Fe(OH)_3_→Fe_2_O_3_ + 3H_2_O(6)

These two kinds of corrosion products (γ-FeOOH, Fe_2_O_3_) are loose and porous, and cannot play a good protective role [32]. With the progress of corrosion, the contact between the corrosion product film and the dissolved oxygen decreases. Under the condition of insufficient dissolved oxygen in corrosion products, γ-FeOOH will generate Fe_3_O_4_ with Fe^2+^, part of which is γ-FeOOH, which will also turn into α-FeOOH [33]. Therefore, α-FeOOH and Fe_3_O_4_ are mainly produced at the contact position between the corrosion product film and the substrate, forming a dense protective inner product film.
6Fe(OH)_2_ + O_2_→2Fe_3_O_4_ + 6H_2_O(7)

The formation of these two corrosion products requires a large amount of Fe^2+^, so it is mainly pitted into the matrix, resulting in many pitting pits on the surface of the matrix [28]. Over time, the yellow corrosion products with large particles on the surface will fall off naturally (show as Figure 7), and some black corrosion products inside will leak out (show as Figure 9).

As shown in Figure 12b, when the flow rate is 0.2 m/s, the flow of seawater is more convenient for the exchange of substances in the solution and the progress of corrosion. Therefore, the formation rate of the corrosion products is greater than that of stripping, and the corrosion products are accumulated (Refer to Figure 6). At this time, the internal corrosion product film is more dense and more protective. The corrosion rate after the corrosion product film is completely formed is smaller than that under static corrosion. However, compared with static corrosion, the loose corrosion products on the surface are easier to be stripped by the impact of seawater and sea sand. At this time, the corrosion product film is mostly α-FeOOH and Fe_3_O_4_ from the inner layer, and the surface is alternating brown yellow and black (Refer to Figure 7).

As shown in Figure 12c, when the flow rate is 0.4 m/s, with the increase in flow rate, the stripping speed of corrosion products is slightly less than the formation rate of corrosion products. At this time, the dissolved oxygen in water is more convenient to have contact with the metal matrix, the corrosion rate is high, and only two corrosion products γ-FeOOH and Fe_2_O_3_ are formed. As shown in Figure 12d, when the flow rate reaches 0.6 m/s, the formation rate of corrosion products is almost equal to the stripping rate. The corrosion products will be peeled off soon after they are generated and form an erosion gully. The composition of the corrosion products is the same as that at 0.4 m/s (refer to Figure 11).

In the process of erosion–corrosion, the flow rate mainly affects the erosion–corrosion mechanism through the stripping degree of the corrosion products. When the flow rate is less than 0.2 m/s, the corrosion products peel less, so the corrosion products accumulate. This phenomenon makes the mass transfer more difficult at a low flow rate, reduces the internal oxygen content, produces the dense corrosion products, and reduces the corrosion rate. When the flow rate is greater than 0.4 m/s, a large number of corrosion products are stripped, and the generation rate of corrosion products is much lower than the stripping rate of the corrosion products. The increase in the flow rate further strengthens the mass transfer, resulting in erosion and enhanced corrosion [13,29,34,35].

## 4. Conclusions

This article conducts a thorough study of the erosion–corrosion behavior of X65 friction stir welded joints under different flow rates in simulated seawater through electrochemical experiments, surface analysis, and corrosion product detection.The results of the electrochemical experiments show that the corrosion product film of the friction stud welded joints in simulated seawater has a certain protective effect on the substrate under static corrosion conditions. Under the coupling effect of corrosion erosion, the corrosion rate of the specimen decreases first and then increases with the increase in flow rate. After 168 h, the self-corrosion current of the specimen at a flow rate of 0.6 m/s is about 2.73 times that of the static corrosion.The surface layer of the corrosion product film of the friction stud welded joint under the static corrosion condition is loose and the inner layer is dense, which has a certain protective effect on the matrix. The corrosion products of the welded joints in a seawater environment are γ-FeOOH, α-FeOOH, Fe_3_O_4_, and Fe_2_O_3_ when the flow rate is less than 0.2 m/s; when the flow rate is greater than 0.4 m/s, the corrosion products affected by erosion are only γ-FeOOH and Fe_2_O_3_.When the flow rate is less than 0.2 m/s, the peel strength of the corrosion products is low, and the accumulation of the corrosion products leads to the formation of a dense corrosion product film in a low oxygen environment, which reduces the corrosion rate. When the flow rate is greater than 0.4 m/s, the peel strength of the corrosion product film increases with the increase in flow rate. A large number of corrosion products were stripped, and further increased the corrosion rate.

## Figures and Tables

**Figure 1 materials-16-04326-f001:**
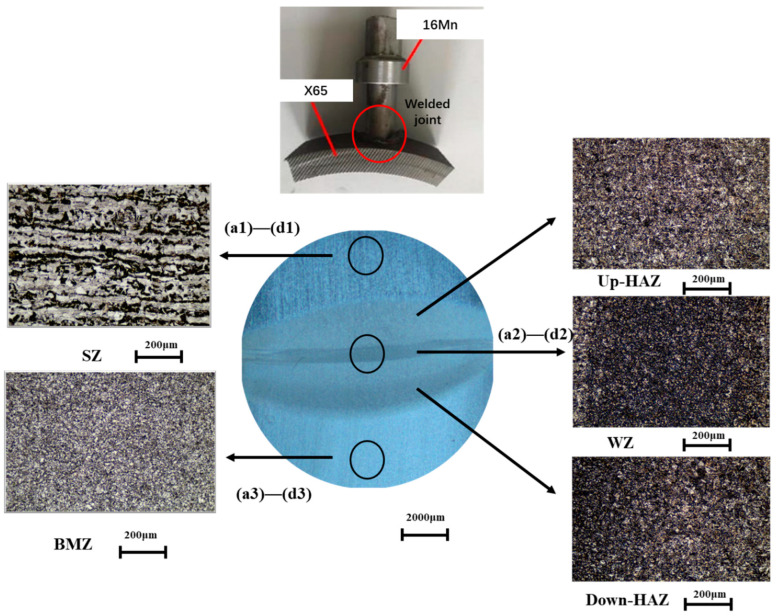
The location of the measuring points for SEM micro corrosion morphology and micro characterization of the five regions of the welded joint.

**Figure 2 materials-16-04326-f002:**
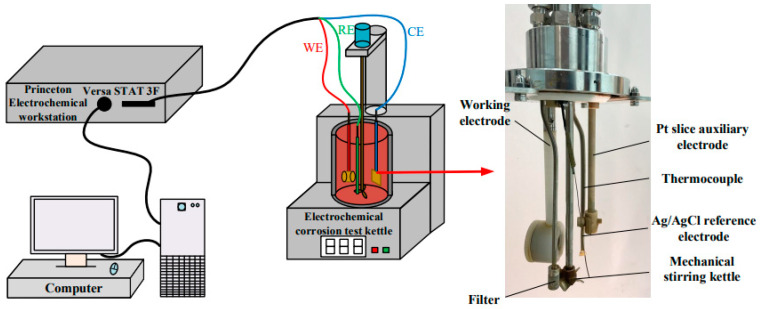
The electrochemical measurement system and electrode distribution.

**Figure 3 materials-16-04326-f003:**
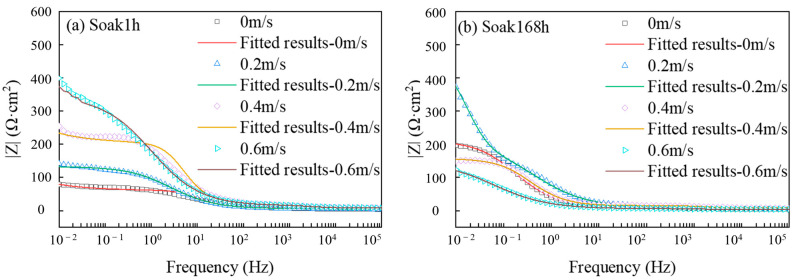
Bode diagram of an X65 FSW joint immersed in a 3.5% NaCl solution for 1 h (**a**) and 168 h (**b**) at different flow rates.

**Figure 4 materials-16-04326-f004:**
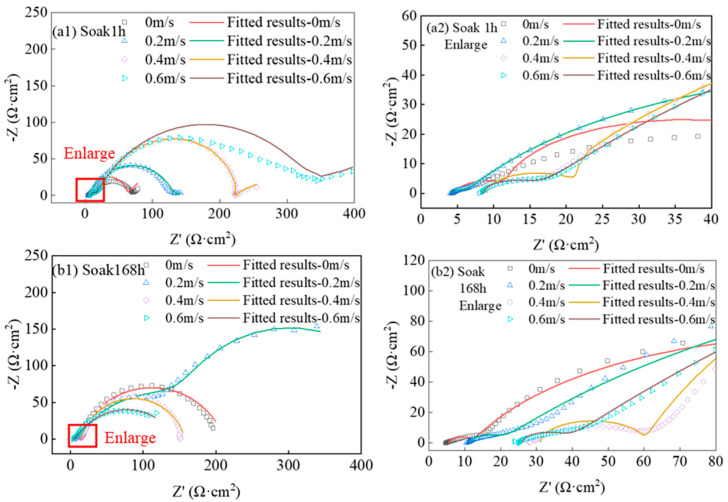
Nyquist diagram and local enlarged diagram of an X65 FSW joint immersed in 3.5% NaCl solution for 1 h (**a1**,**a2**) and 168 h (**b1**,**b2**) at different flow rates under simulated real working conditions.

**Figure 5 materials-16-04326-f005:**
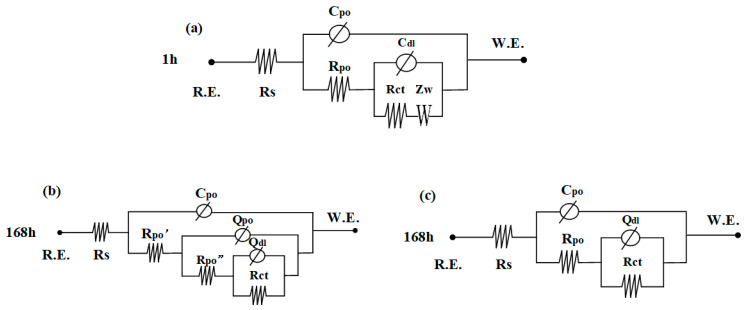
Equivalent circuit of the test piece under different flow rates (**a**) test pieces for 1 h when the flow rate is 0/0.2/0.4/0.6 m/s, (**b**) test pieces for 168 h when the flow rate is 0.2 m/s, (**c**) test pieces for 168 h when the flow rate is 0/0.4/0.6 m/s.

**Figure 6 materials-16-04326-f006:**
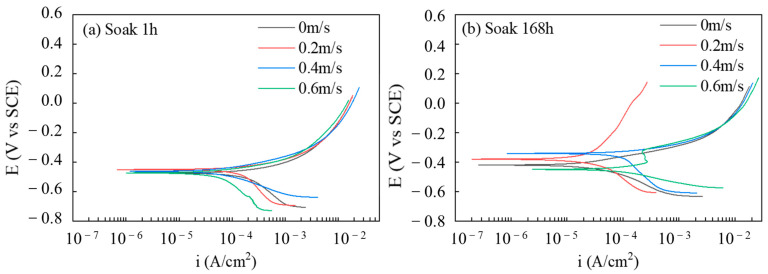
Tafel of the X65 FSW joint soaked in 3.5% NaCl solution at different flow rates for 1 h (**a**) and 168 h (**b**).

**Figure 7 materials-16-04326-f007:**
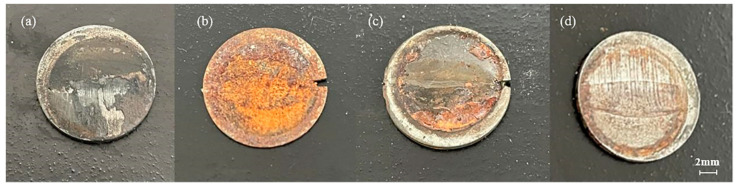
The X65 FSW joints are at 0 m/s (**a**), 0.2 m/s (**b**), 0.4 m/s (**c**), and 0.6 m/s (**d**). Macro morphology after 168 h in 3.5% NaCl solution.

**Figure 8 materials-16-04326-f008:**
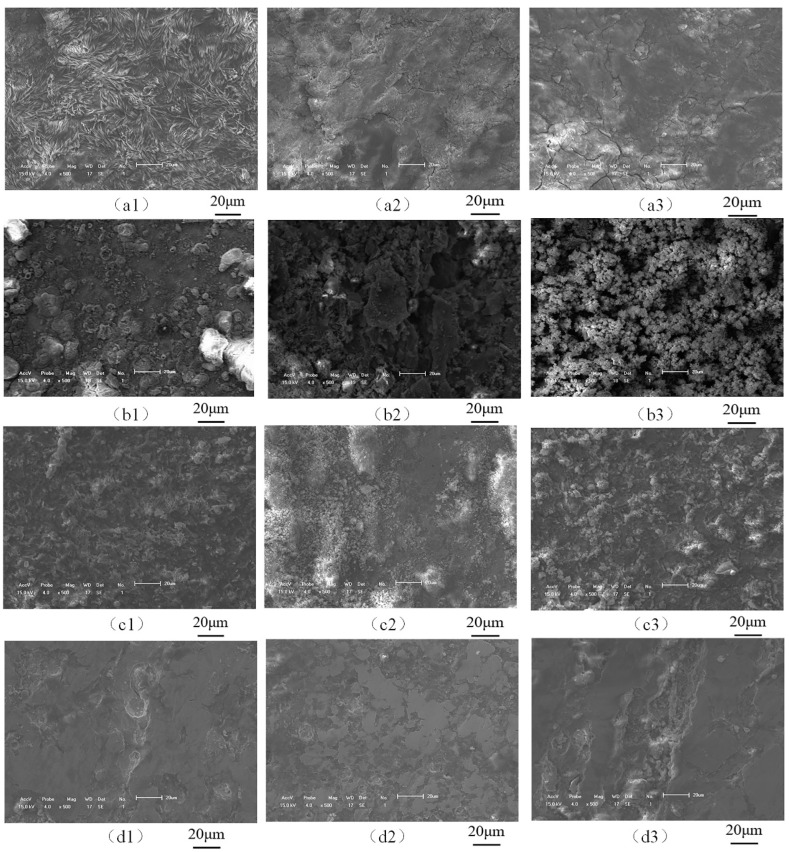
Microscopic morphology of the X65 FSW joints after 168 h in a 3.5% NaCl solution at flow rates of 0 m/s (**a1**–**a3**), 0.2 m/s (**b1**–**b3**), 0.4 m/s (**c1**–**c3**), and 0.6 m/s (**d1**–**d3**).

**Figure 9 materials-16-04326-f009:**
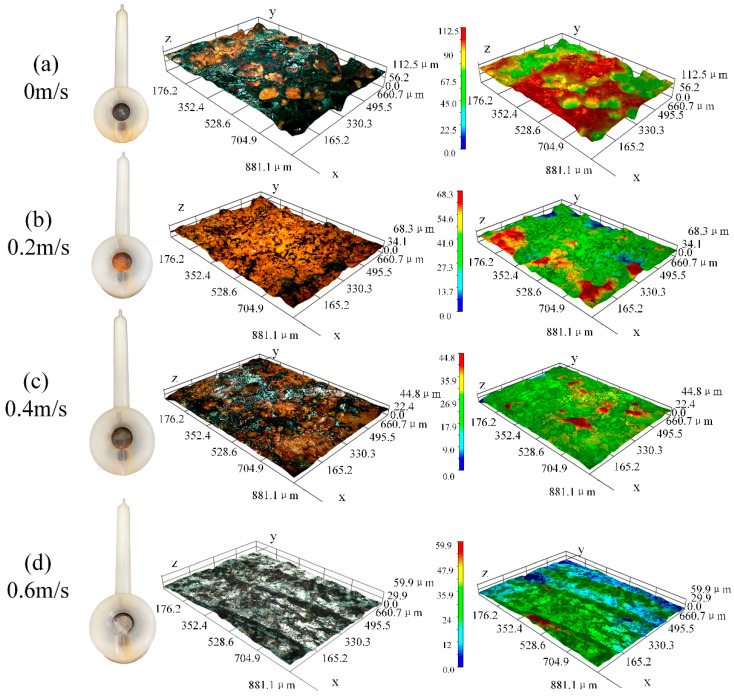
The X65 FSW joints are at 0 m/s (**a**), 0.2 m/s (**b**), 0.4 m/s (**c**), and 0.6 m/s (**d**) depth of field morphology after 168 h in a 3.5% NaCl solution.

**Figure 10 materials-16-04326-f010:**
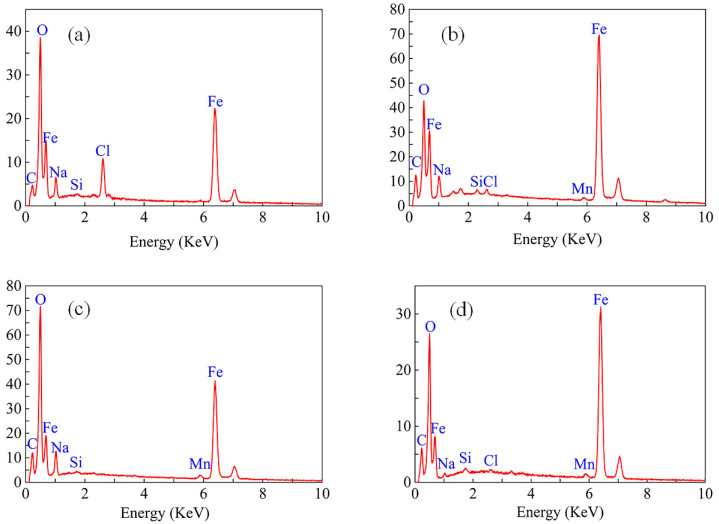
XRD of the X65 friction stud welded joint immersed in a 3.5% NaCl solution for 168 h at the flow rates of 0 m/s (**a**), 0.2 m/s (**b**), 0.4 m/s (**c**), and 0.6 m/s (**d**).

**Figure 11 materials-16-04326-f011:**
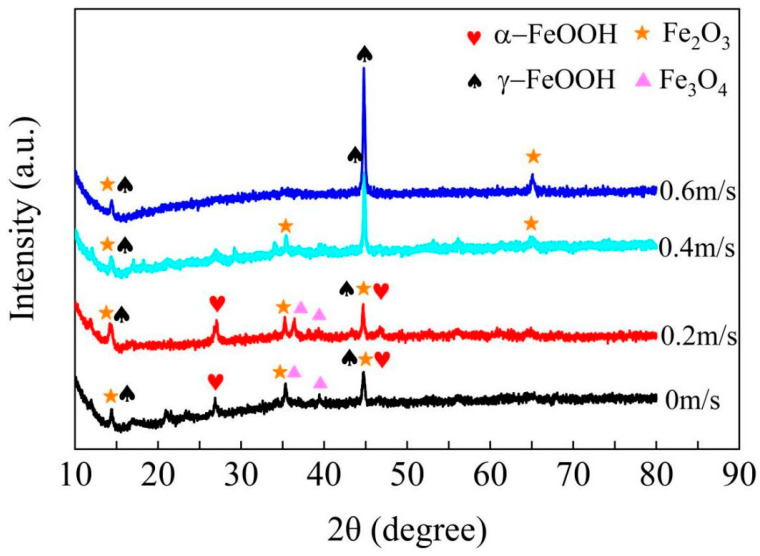
XRD of the X65 friction stud welded joint immersed in a 3.5% NaCl solution for 168 h at the flow rates of 0 m/s, 0.2 m/s, 0.4 m/s, and 0.6 m/s.

**Figure 12 materials-16-04326-f012:**
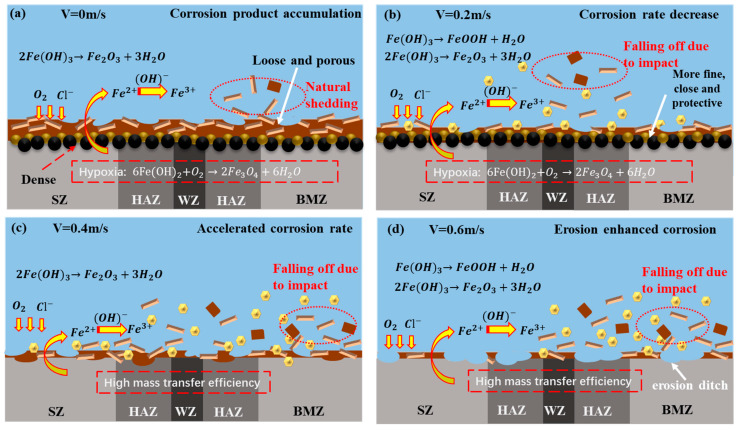
Erosion mechanism diagram under different seawater velocities.

**Table 1 materials-16-04326-t001:** EIS fitting results at different flow rates.

Component	1 h				168 h			
	0 m/s	0.2 m/s	0.4 m/s	0.6 m/s	0 m/s	0.2 m/s	0.4 m/s	0.6 m/s
R_s_ (Ω·cm^2^)	4.914	5.00	9.06	8.56	5.42	5. 62	7.88	4.33
C_po_ (μF/cm^2^)	42.41	80.07	11.88	7.67	117.0	22.45	13.29	120.9
R_po_ (Ω·cm^2^)	8.28	8.80	13.64	7.88	7.28	-	7.24	2.15
C_dl_ (mF/cm^2^)	3.87	4.25	2.30	1.11	-	-	-	-
R_ct_ (Ω·cm^2^)	57.58	116.68	200.9	336.4	199.3	196.3	142.6	141.0
Q_dl_ (mF/cm^2^)	-	-	-	-	6.20	3.23	4.74	16.27
R_po_′ (Ω·cm^2^)	-	-	-	-	-	4.582	-	-
Q_po_ (mF/cm^2^)	-	-	-	-	-	32.81	-	-
R_po_″ (Ω·cm^2^)	-	-	-	-	-	168.9	-	-
Z_w_ (Ω·cm^2^)	0.170	0.12	0.10	0.16	-	-	-	-

**Table 2 materials-16-04326-t002:** The polarization curve fitting results at different flow rates.

Parameter	1 h				168 h			
	0 m/s	0.2 m/s	0.4 m/s	0.6 m/s	0 m/s	0.2 m/s	0.4 m/s	0.6 m/s
I_corr_ (μA/cm^2^)	154.24	120.31	43.87	35.42	30.81	20.39	69.16	84.22

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
