# Peer review of "Erosion–Corrosion Behavior of Friction Stud Welded Joints of X65 Pipelines in Simulated Seawater under Different Flow Rates"

_materials, 2023, doi:10.3390/ma16124326_

Round 1

Reviewer 1 Report

The author must improve the quality of the presentation. There are a lot of mistakes in Equation numbering and reference, etc. 

There are a lot of comments and suggestions to be done in the in the pdf file

Author Response

Thank you very much for your valuable comments on our manuscript. Your suggestions have played a crucial role in helping us to further improve the quality of our article. After carefully considering your feedback, we have carried out appropriate modifications and enhancements, which we outline in our response below.

Reviewer 2 Report

1. I recommend correcting the title of the article, “Erosion-corrosion behavior of friction stud welded joints of Ð¥65 steel in simulated seawater under different flow rates”.

 2.  If possible, please, improve the quality of the photo of the microstructure in Fig. 1. The quality of all figures in the article should be improved, if possible.

3.  Explain the designations a1,2,3, b1,2,3, c1,2,3, d1,2,3 in Fig. 1.

4.  Provide the welding mode, or at least give a reference to the work in which it is described.

5. To justify, in accordance with which regulatory document, such flow rates were chosen.

6. Since a sample of a welded joint is being considered, it is necessary to note how the corrosion products are located on the surface, which part of them is in the weld area, and which is on the base metal. If there is no difference, it must be said.

7.  References numbered 4,5, 10,11,17, 26, 27, 33, 35 have not related to the topic of the article.

Author Response

Thank you very much for your valuable comments on our manuscript. Your suggestions have played a crucial role in helping us to further improve the quality of our article. After carefully considering your feedback, we have carried out appropriate modifications and enhancements, which we outline in our response below.

Thank you for your professional criticism and suggestions, which have played an irreplaceable role in promoting the argumentation, academic exploration, and in-depth analysis of this article. We will do our best to meet the requirements of the journal and international standards, and look forward to receiving your careful review and valuable feedback again.

Reviewer 3 Report

The manuscript deals with erosion-corrosion behavior of X65 friction stud welded joint. The following are my comments. 1. Some of the literature are not directly relevant to the study proposed. 2. Specify the sea sand mean diameter, composition etc. as it affects the erosion-corrosion behavior. 3. Though the experiments are conducted with NaCl + sea sand, in many places only NaCl is alone quoted. Check for consistency. 4. The quality of all the figures are very poor. Provide >300dpi image. 5. Throughout the text we can find Error! Reference source not found. Do necessary correction. 6. How NaCl and sea sand are mixed and proof to confirm their homogeneity. 7. Explain with labels in Figure 8.  8. Explanation about Figure 9 is poor. Need to discuss the technical part rather than telling the color. 9. EDS analysis is missing in the paper and only Table 3 is included. Preferably remove Table 3 and provide EDS. How the readers shall ensure your result quality? 10. Language has to be improved.

Language is very poor and need special care.

Author Response

(The authors gave the same response as above.)

Reviewer 4 Report

Comments for materials-2385023:

The paper “Erosion-corrosion behavior of friction stud welded joints of pipelines in simulated seawater under different flow rates” investigated the effect of effects of corrosion and erosion-corrosion at different flow rates on friction stud welded joints in simulated seawater. Results indicate that the corrosion current density decreased first and then increased with the increase of simulated seawater flow rate. In general, this paper is clearly written. The title is adequate and appropriate for the content of the article, and the abstract is complete and suitable for inclusion by itself in an abstracting service. The conclusion represents adequately the results and main contributions of the article. But the manuscript in the current version cannot be accepted for publication. I recommend that this article be minor revised. Some comments are as follows:

1.  Is the solution for electrochemical tests deaerated or aerated? What’s the DO content in the solution? Because oxygen has a significant effect on the corrosion behavior of materials, it is better to describe whether the solution used in this study is aerated or deaerated.

3.  Fig. 4, the Nyquist plots should be modified. The X-axis and Y-axis should be in the same range. And the maximum and minimum frequency of the data point should be labeled in the plot.

4.  All the figure numbers in the manuscript are errors. Please correct them.

5. Fig. 8, the quality of the SEM images is not good enough. It is better to improve the SEM quality.

6. What’s the relationship between the flow rate and corrosion rate? Generally, people believe that the corrosion products can be removed easily and the fresh surface of the material can contact the solution directly at a high flow rate, resulting high corrosion rate.

Considering all the problems mentioned here above, some additional evidence should be added to support the statement in the manuscript, and some parts of this manuscript should be revised. I recommend that the manuscript should be minor revised.

Minor editing of English language required. It is better to have the English of the manuscript polished by native speakers.

Author Response

(The authors gave the same response as above.)

Round 2

Reviewer 1 Report

The reference numbering must be verify

Reviewer 3 Report

The manuscript shall be accepted